# Response of Extremely Small Populations to Climate Change—A Case of *Trachycarpus nanus* in Yunnan, China

**DOI:** 10.3390/biology13040240

**Published:** 2024-04-05

**Authors:** Xiaofan Wang, Xuhong Wang, Yun Li, Changhao Wu, Biao Zhao, Mingchun Peng, Wen Chen, Chongyun Wang

**Affiliations:** 1Institute of Ecology and Geobotany, Yunnan University, Kunming 650504, China; xfwangynu@163.com (X.W.); 15240801598@163.com (Y.L.); 18213535564@163.com (B.Z.); mchpeng@ynu.edu.cn (M.P.); 2College of Ecology and Environment, Yunnan University, Kunming 650504, China; 18946849834@163.com; 3Southwest United Graduate School, Yunnan University, Kunming 650092, China; wuchanghaochn@gmail.com (C.W.); chenwen-dq@ynu.edu.cn (W.C.)

**Keywords:** *Trachycarpus nanus*, MaxEnt model, potential distribution area, climate change, ENMeval

## Abstract

**Simple Summary:**

*Trachycarpus nanus* is a national, second-class, rare and endangered plant in China, and a plant species endemic to Yunnan. This species has a high medicinal, ecological, and scientific value, but it is currently on the verge of extinction. Therefore, it is urgent to learn its extinction risk under global warming. We predict its multi-temporal distribution pattern based on the optimized MaxEnt model. The results show that precipitation is the most important factor. In the future, drastic climatic changes and human disturbances may lead to the extinction of this species, so it is necessary to strengthen the protection of this species using specific strategies.

**Abstract:**

Climate change affects the geographical distribution of plant species. Rare *Trachycarpus nanus* with a narrow distribution range, high medicinal value and extremely small population is facing increasing extinction risks under global climate change. In this study, 96 recorded occurrences and 23 environmental factors are used to predict the potential suitable area of *T. nanus* based on the optimized MaxEnt (3.4.4) model and ArcGIS (10.7) software. The results show that when the parameters are FC = LQ and RM = 1, the MaxEnt model is optimal and AUC = 0.946. The distribution patterns were predicted in the past, present, and four future phases, i.e., 2021–2040 (2030), 2041–2060 (2050), 2061–2080 (2070), and 2081–2100 (2090). The main factors are the annual precipitation (bio12), mean temperature of the coldest quarter (bio11), temperature seasonality (bio4), precipitation of the wettest quarter (bio16), and isothermality (bio3). The potential distribution of *T. nanus* is primarily concentrated in central Chuxiong, encompassing a total potential suitable area of 5.65 × 10^4^ km^2^. In historical periods, the total habitat area is smaller than that in the present. In the future, the potential suitable area is generally increased. The centroid analysis shows that *T. nanus* will move to a high-altitude area and to the southeast. But its dispersal capacity may not keep up with the climate change rate. Therefore, additional protection sites for this species should be appropriately established and the habitat connectivity should be enhanced.

## 1. Introduction

The geographical distribution pattern and population dynamics of plant species are closely related to the climate pattern and are driven by climate shifts [1,2,3]. Plants continuously respond to historical climate oscillations [4,5]. The Sixth Assessment Report (AR6) of the Intergovernmental Panel on Climate Change (IPCC) states that the surface temperature in the last decade has increased by 1.53 ± 0.15 °C compared to 1950–1990, and surface temperatures will continue to rise in the future [6]. The already discernible climate-forced changes in animal and plant populations caused by global warming will ultimately produce possible extinctions [7,8]. And individual species vary greatly in their range shift depending on internal species traits and external drivers of change [9,10,11]. In the process of a plant’s long-distance dispersal, the population may become extinct before finding a highly suitable habitat in the future. If plants can quickly spread to suitable habitats in the future, the risk of plant habitat change and population extinction will be greatly reduced [12]. Therefore, predicting the potential suitable distribution and change rate of plant species is of great significance for evaluating their extinction risk and for the policy-making of effective conservation [13].

Species distribution models (SDMs) are based on niche theory to analyze the correlation between species occurrences and environmental factors, then the results are projected in geographical space to predict the ecologically potentially suitable area of the species [14]. At present, the MaxEnt model, Climex model, Bioclim model, Genetic Algorithm for Rule-set Prediction and so on are the most used ecological niche models [15,16,17]. The MaxEnt model has the advantages of having a simple operation, high prediction accuracy, and strong explanatory power [18,19]. Therefore, it is widely used in various research fields, especially applied to endangered and cherished plant species whose niche distribution is restricted by environmental factors and human activities [20,21,22,23,24].

Plant species with extremely small populations (PSESPs) are distributed in a narrow area and their population continues to decrease due to the influence of biotic or abiotic factors. Existing populations have fallen below the minimum survival limit of stable survival, and it is difficult to maintain the normal reproduction and renewal of wild plants, which are at risk of extinction at any time [25]. At present, the investigation of wild plants with extremely small populations mainly focuses on their field resource distribution, community, breeding system, conservation genetics, and conservation methods and strategies [26,27]. However, PSESPs are facing more extinction threats under intense climate change. *Trachycarpus nanus* Beccari is *Trachycarpus* H. Wendl of Arecaceae Bercht. and J. Presl. *T. nanus* is a perennial shrub without aboveground stems, it is dioecious, has leaves clustered in the ground surface, and a palmate that is deeply divided [28,29,30]. Its underground stem is like a coiled dragon, so it is named “dragon palm” in Chinese (Figure 1). *T. nanus* is an endemic species in Yunnan, China, mainly distributed in Chuxiong Prefecture and the adjacent areas (Figure 2). It usually grows in small populations and isolated scatters in mountain areas at an altitude of 1800–2300 m (Appendix A). In the past few decades, the populations of *T. nanus* have decreased sharply [31]. The distribution area of *T. nanus* is densely populated and developed earlier, and deforestation and agricultural land demand lead to its habitat loss [32]. *T. nanus* has medicinal usages, as the flowers can cure nephropathy, the seeds can cure dizziness and headaches, and the leaf sheaths can be used as astringent and hemostatic drugs. Additionally, the leaves of the *T. nanus* are made into brooms by local people, which are durable and cheap. Its medicinal value and economic value make excessive use of its resources, so that the *T. nanus*, which was originally common in mountainous areas of Chuxiong Prefecture, is hard to find today [28,32,33]. The development zones with good wind energy resources in Yunnan Province cover most areas where the *T. nanus* is distributed [34]. Wind power farm construction is a threat to its habitats [35,36]. In addition, livestock overgrazing (Appendix A) [29] and road construction [28] are also human activities that threaten *T. nanus*. According to field investigations and the literature, the invasive species *Ageratina adenophora* has occupied a large number of *T. nanus*’ habitats (Appendix A), seriously affecting the survival and regeneration of its seedlings [37].

*T. nanus* is listed as endangered on the IUCN Red List (IUCN 2019; http://www.iucnredlist.org, accessed on 4 April 2023), and is under protection as an endangered species Class II in China [33,38]. Its biological characteristics [28], seed germination properties [39], population distribution and endangered status [31] have been studied in past decades. In recent years, the whole chloroplast genome of this species has been sequenced, and it was found that its genetic diversity level is low and it has a close phylogenetic relationship to *Chamaerops humilis*, which distributes in Mediterranean shore areas [40]. Severe climate change has had a wide range of impacts on biodiversity and ecosystem functions [41], especially for endangered plants such as *T. nanus*. In order to prevent its population from further decreasing, it is urgent we protect its existing wild individuals and its population’s habitat. However, there is no in-depth study on how to expand global carbon neutrality to reduce the impact of climate change on *T. nanus*. This study aims to fill this knowledge gap. By applying species distribution models (SDMs) to *T. nanus*, the potential impact of climate change on its living environment is determined, and the corresponding protection strategies are put forward. This will provide the basis for the protection implementation, population recovery and reestablishment of the *T. nanus*. Based on the MaxEnt model and ArcGIS V10.7 software, the spatial changes of *T. nanus*’ distribution are simulated from the last glacial period, i.e., the last interglacial period (LIG, ~120,000–140,000 years BP), the last glacial maximum period (LGM, about 22,000 years ago), and the middle Holocene (MH, about 6000 years ago), to the present period, and into the future in 2030, 2050, 2070, and 2090. We want to understand (1) the key environmental factors affecting the distribution of *T. nanus*, (2) what differences of the spatial pattern and change rate of the *T. nanus* have been driven by climatic change in the past and will be in the future, (3) the threats to its population’s survivability from human-driven climate change, and to put forward corresponding protection strategies.

## 2. Materials and Methods

### 2.1. Species Distribution Data

The distributions of *T. nanus* were collected from the Global Biodiversity Information Data Center (https://www.gbif.org/, accessed on 24 March 2023) and the Chinese Virtual Herbarium (https://www.cvh.ac.an/, accessed on 24 March 2023), as well as via field survey. Artificial transplanted sites were removed from natural distribution record points. In total, there are 101 *T. nanus* distribution records and 25 field samples. Firstly, by observing the spatial distribution of the *T. nanus* distribution points, it is found that some recording points are in an aggregation state and there is a problem of spatial autocorrelation, which may be caused by sampling deviation or the limitations of data collection methods. When there are local spatial clusters, the modelling is usually over-fitted (reducing the prediction ability of the model) and exaggerating the performance value of the model [42]. Based on the SDMtools toolbox (http://www.sdmtoolbox.org/, accessed on 20 April 2023), we only keep one distributed data point in the range of 10 km × 10 km, reducing the occurrence position to a single point within the distance, and achieving the situation where the whole single grid meets the requirements [24,43]. Finally, 96 distribution points were used in the modeling (Figure 2). Distribution data with latitude and longitude were saved in “csv” format for later use.

### 2.2. Environment Data

Environmental data included 19 bioclimatic factors, 3 topographic factors and the land use type. Bioclimatic data for three different periods (past, current, and future) were obtained from the WorldClim database (https://worldclim.org/, accessed on 13 April 2023). Past climatic periods include the LIG, LGM and MH. Current climate data covers the 1970s to the 2000s. Future climate data includes four periods: the 2030s (2021–2040), the 2050s (2041–2060), the 2070s (2061–2080), and the 2090s (2081–2100). The future climate data from the Coupled Model Inter-comparison Project Phase 6 (CMIP6) were selected [44], and the model was the BCC-CSM2-MR (the Beijing Climate Center Climate System Model) [45]. Meanwhile, we chose three different, shared socio-economic pathways: SSP1-2.6, SSP2-4.5, and SSP5-8.5 [46]. They represent low radiative forcing, medium radiative forcing, and high radiative forcing, respectively. In addition, the DEM data came from the Geospatial Data Cloud (https://www.gscloud.cn/, accessed on 14 April 2023). And terrain data (elevation above sea level, aspect, and slope) were derived from the DEM. The land use data came from the National Earth System Science Data Center (https://www.geodata.cn/, accessed on 14 April 2023). The nature reserves data came from the GEOVIS Earth Open Platform (https://www.geovisearth.com/, accessed on 15 April 2023). Spatial resolution should be selected according to the ecological and biological characteristics of the plant species. Variables affect the distribution of the species on relevant scales, which depends on the geographical scope and granularity of the modeling tasks [47]. On one hand, if the spatial resolution of the environmental factors is too coarse, some environmental factors may be integrated, which would indicate it is unable to reflect the real influences on *T. nanus*. On the other hand, constraining by suitable habitats and seed dispersal, *T. nanus* has small and scattered populations in its geographical range [39]. Environmental variation in the geographical range could be well indicated by 1 km resolution variables. And, the 1 km resolution bioclimatic data of WorldClim is accessible and is a rational choice. Therefore, we designated the spatial resolution of all data as 1 km. The above environmental variables were converted to an “asc” format based on ArcGIS software.

Due to the multicollinearity among various environmental factors causing overfitting, the correlation analysis was carried out using R software (4.1.3) for each environmental factor [48,49,50] (Figure 3). Meanwhile, based on the preliminary simulation results of the MaxEnt model (Table 1), environmental factors with a correlation ≥ 0.8 and a low contribution rate were eliminated [23,43,46]. Finally, 12 environmental factors were selected to predict the potentially suitable area of the *T. nanus* (Table 2).

### 2.3. Model Setting and Evaluation

The choice of the MaxEnt model needs to consider the particularity of the research problem and the characteristics of the data. The particularity of the research problem is that the (1) MaxEnt model has a good prediction ability in the study of endangered species [23,43], and that the (2) MaxEnt model can be used to predict the current state of species distribution, future scenarios, niche expansion, and other different forecasting purposes. This study not only predicts the current distribution, but also that of the past and future. Data characteristics include: (1) In this study, the *T. nanus* has a small population, and the data amount is not much. The MaxEnt model has a strong processing ability for small sample datasets, so it is suitable for this case with a small sample size [14]. (2) Environmental factors such as the isothermality and land use type are discrete variables, while the precipitation is a continuous variable. The MaxEnt model can integrate various types of environmental factor data, including continuous variables and classified variables, and can include the interaction of different types of variables [47]. (3) *T. nanus* is a rare and endangered species with a narrow distribution. A reliable prediction could be produced by the MaxEnt model for discrete species distribution data, so it is suitable for dealing with an uneven distribution or locally rare species [23]. Compared to other models, the MaxEnt model has the best explanatory power and the most accurate prediction performance [18,19], a concise mathematical definition [14], a capacity for avoiding the sampling bias problem [51], and a suitability for the conservation planning algorithm [42]. Therefore, we chose this model to predict the suitable area of the endangered plant *T. nanus*.

The ENMeval software package (R 4.1.3) was used to optimize the MaxEnt model [52]. In the MaxEnt model, the regularization multiplier (RM) is an adjustment parameter to balance the fitting degree and complexity of the model. The greater the regularization multiplier, the greater the penalty for the complexity of the model, making it more inclined to choose a simple model; the smaller the regularization multiplier, the smaller the penalty for model complexity, and the more prone to over-fitting. Feature combination (FC) refers to combining different environmental factors to form new features in order to increase the model’s ability to explain species distribution. Through feature combination, the interaction and nonlinear relationship between environmental factors can be better captured, thus, it improves the fitting ability and prediction performance of the model. In practical application, it is very important to choose the appropriate regularization multiplier and feature combination, which need to be determined via cross-validation and other methods. We set up 12 kinds of regularized multipliers, with values ranging from 0.5 to 6, and combined them with 6 kinds of feature combinations, i.e., L, LQ, LQH, H, LQHP, and LQHPT (L—linear, T—threshold, Q—quadratic, H—hinge, and P—product), for parameter cross-combination [53]. Then, the parameters of the above 72 combinations were tested in the ENMeval package. Finally, the best parameter combination (delt. AICc = 0) was selected for MaxEnt modeling [54].

We put 96 occurrences of *T. nanus* and 12 filtered environmental factors into the MaxEnt software (version 3.4.4), and set the maximum number of iterations to 10,000. The jackknife method was used to determine the importance of each variable. The output format was set as “Logistic”. In total, 75% of the data were used as training data and 25% as detection data [12,55]. The model was repeated 10 times, and the optimized RM and FC values were then selected for operational modeling. The AUC value under the receiver operating characteristic curve (ROC) in the MaxEnt model output was used to test the accuracy [56]. The value of the AUC ranged from 0–1 [57,58]. If the value is larger, the reliability of the model is higher, as well as the effect being better [59,60]. A value of the AUC that is less than 0.7 is a poor prediction; 0.7–0.8, mediocre prediction; 0.8–0.9, good prediction; more than 0.9, the prediction is excellent [61,62].

### 2.4. Classification of Potentially Suitable Area

The average value of the MaxEnt model output results was imported into ArcGIS, and the “asc” format file was converted into raster data using the conversion tool [63]. The natural breaks (Jenks) method was used to reclassify the results of the simulated *T. nanus*’ suitable area, which was divided into four suitable levels: an unsuitable area (0–0.2), a less suitable area (0.2–0.4), a medially suitable area (0.4–0.6), and a highly suitable area (0.6–1) [64,65]. The projection coordinate system chosen in this study was the WGS 1984 UTM Zone 48N and the proportion and area of each level of the suitability area were calculated.

### 2.5. Spatial Pattern Change and Centroid Analysis of Species’ Potentially Suitable Area

Non-suitable/suitable binary matrixes of each period were produced according to the suitable levels. The area of <0.4 was set as the uninhabitable area and assigned a value of 0; the region with a distribution probability ≥0.4 was set as a suitable area, and assigned a value of 1. Through the overlapping of the different period’s binary matrix, 0–0 was defined as the unsuitable area, 0–1 as an increase in the suitable area, 1–0 as a loss in the suitable area, and 1–1 as the reserved suitable area [66]. Next, the variation of the suitable area was calculated to learn the geographical contraction, retention, and expansion in different periods.

The centroid analysis was applied by the SDM toolbox. Based on the above non-suitable/suitable binary matrix, the centroid changes of the potentially suitable area in different periods were calculated and simulated.

## 3. Results

### 3.1. Accuracy Evaluation of the MaxEnt Model

The model optimized by the ENMeval package was the combination of FC = LQ and RM = 1, i.e., delta. AICc = 0. The optimized AICc and the difference in the AUC between the training dataset and the test dataset are lower than the default setting (Figure 4, Table 3). Using the ROC curve analysis method to test the distribution results predicted by the MaxEnt software (version 3.4.4), the AUC value, the area under ROC curve, is the accuracy index of the prediction ability of the model. The results show that the AUC value is 0.946 (Figure 5), and the simulation effect of the model has reached a high level, indicating that the experimental results are highly reliable.

### 3.2. Dominant Environmental Factors Affecting the Distribution of T. nanus

The MaxEnt model weights or integrates environmental factors into the model mainly through the following steps [14]: (1) Select the environmental factors that may affect species distribution based on ecological knowledge, field investigation or literature review. (2) Obtain GIS data of the selected environmental factors, which are usually provided in a grid or vector format and cover the spatial scope of the study area. (3) Weight the environmental factors based on the correlation between the environmental factors and the known species distribution. The MaxEnt model can track environmental variables with a high contribution rate, gradually modify a single factor to increase the gain value, and assign the gain value to the environmental variables on which the factor depends, which are expressed as a percentage [67]. The importance of permutation is determined by the random substitution value of the environmental variables at training points and the decrease in the AUC value. The decrease in the AUC value indicates that the variable depends on the model [68]. The jackknife test uses one variable in turn or excludes one variable in turn to build models, and compares the differences of the training gain, test gain, and the AUC value between models to evaluate the importance of the environmental variables [69]. (4) Model training: the MaxEnt model uses the known species distribution points and the corresponding environmental factor data for training. In the training process, the model will learn the relationship between the environmental factors and the species distribution, and optimize the model parameters to fit the known data to the greatest extent. (5) Predicting species distribution: the trained MaxEnt model can be used to predict the species distribution in unknown places. In the process of prediction, the model will calculate the suitability of each site for species survival according to the data of the environmental factors and the learned weights. Dominant environmental factors affecting *T. nanus*’ ecological distribution were revealed via the contribution rate, permutation contribution rate, and the single factor response curve in a jackknife analysis (Table 2, Figure 6 and Figure 7). The environmental factors with a contribution rate above 5% are bio12 (48.9%), bio11 (11.3%), bio4 (9.6%), bio16 (8.0%), and bio3 (7.1%). These five environmental factors account for a contribution rate of 84.9%, which is important for the distribution of *T. nanus*.

In order to clarify the relationship between the dominant environmental factors and the existence probability of the *T. nanus*, single factor response modeling was conducted based on the above five environmental factors. It is generally believed that an existence probability greater than 0.5 can be regarded as an optimal survival range of a species [70]. This is illustrated in the single factor response curve (Figure 7). When the annual precipitation (bio12) is greater than 407 mm, its existence probability increases, and it reaches its peak at 667 mm then drops with the continuous increase of bio12. When bio12 reaches 1200 mm, the probability of existence is almost 0. The optimal survival range of bio12 is 407–844 mm. The existence probability of *T. nanus* also increased with the mean temperature of the coldest quarter (bio11), the seasonal variation coefficient of the temperature (bio4), the precipitation of the wettest quarter (bio16), and the isothermality (bio3). And, after reaching the optimal survival peak, it then decreases with these environmental factors. The optimal peak values for these factors are 7.7 °C (bio11), 454.8 (bio4), 432.3 mm (bio16), and 48.2 (bio3). As for their optimal survival ranges, bio11’s is 7–10 °C, bio4’s is 424–484, bio16’s is 401–467 mm, and bio3’s is 46–48.

### 3.3. Present Potentially Suitable Area for T. nanus

At present, the potentially suitable area for *T. nanus* is mainly concentrated in central Yunnan, with a total suitable area of 5.65 × 10^4^ km^2^, accounting for 14.35% of the total land area of Yunnan Province. The highly suitable area is about 1.00 × 10^4^ km^2^, accounting for 2.54% of the area of Yunnan Province; the medially suitable area is about 1.79 × 10^4^ km^2^, accounting for 4.54%; and the less suitable area is about 2.86 × 10^4^ km^2^, accounting for 7.26% (Table 4). The proportion of the highly and medially suitable areas of *T. nanus* are small and concentrated. And the proportion of the less suitable area is large and scattered. It is shown that the highly and medially suitable areas are mainly distributed in eastern Dali, southeast of Lijiang and central to Chuxiong. The less suitable area is mainly scattered around the surrounding areas of the highly and medially suitable areas, it reaches Kunming and the northern part of Yuxi (Figure 8).

### 3.4. Potentially Suitable Area for T. nanus under Past and Future Climate Scenario

Historical and future distribution ranges of potentially suitable areas for *T. nanus* change in various ways (Figure 9 and Table 4). From the LIG to the MH, the total potentially suitable area first increases and then drops, and reaches its maximum during the LGM, which is 5.61 × 10^4^ km^2^. In the future, the potentially suitable areas for *T. nanus* show different changes. In the SSP1-2.6 scenario, from 2030 to 2090, the total potentially suitable area will increase first and then decrease. In 2030 and 2070, the potential habitat area will increase by 4.6% and 2.8%, respectively. However, the potential habitat area in 2050 and 2090 will decrease by 3.5% and 3.0%. In the SSP2-4.5 scenario, except during an increase in 2050, the potential habitat area will decrease in other periods. In 2050, it will increase by 2.5%, while in other periods, it will decrease by 9.7%, 13.8%, and 8.8%. In the SSP5-8.5 scenario, it will be lower in 2030 and 2050 than in the current period, which will be a drop of 6.4% and 0.7%. Overall, in the SSP1-2.6 scenario and in the SSP5-8.5 scenario, the area shows a trend of gradual increase, in the SSP2-4.5 scenario, the area shows a trend of gradual decrease. It will increase by 14.9% and 25.3% in 2030 and 2050.

It is indicated in Figure 9 that the potentially suitable area for *T. nanus* has a dissimilar change in each climate period. However, its highly suitable distribution is concentrated in Chuxiong and western Dali. The less suitable habitat has much more spatial fluctuation.

### 3.5. Spatial Variation Pattern of Potentially Suitable Areas for T. nanus

In MH, the change rate is greater, up to 5.66%, but the loss of area is only 0.45 × 10^4^ km^2^ (Table 5 and Figure 10). The loss area is mainly northern and central to Chuxiong and the northeast of Dali, while the increasing area is mainly in eastern Dali, southern Lijiang, and western Kunming. The change is presumed to be due to the great climate alternation in MH. The area increased in the LIG period and the LGM period is similar to that in the MH, but the area lost in the LIG is mainly concentrated in eastern Dali. In the LGM, it is mainly concentrated in the northern part of Chuxiong.

In the SSP1-2.6 scenario, the increased area is the largest and the loss area is the smallest in 2070, the change rate is 4.42%. In 2090, the area increment is the smallest, and the area of loss is less different from that of other periods, making it the smallest change rate, only 1.42%. In general, the distribution of increased area and loss area in the four periods is similar, and the increased areas are mainly concentrated in central and northern Chuxiong and eastern Dali. The loss areas are mainly concentrated in southern Lijiang and northeastern Dali.

In the SSP2-4.5 scenario, the loss area is larger than the area of increase, except in 2030. And the loss area is gradually increasing. The loss area in 2030 and 2050 are relatively dispersed, while they are larger and more concentrated in 2070 and 2090. And the main loss areas are located in southeastern Lijiang, northeastern Dali, and eastern Kunming.

In the SSP5-8.5 scenario, the increase area is the largest in 2090, the area change rate is up to 17.7%. The distribution is expanded to the southeast, mainly increasing in southern Kunming and northern Yuxi.

### 3.6. Centroid Shifts in the Potentially Suitable Areas for T. nanus during Different Periods

The centroid of the potentially suitable area is explored to illustrated the changing direction and range change of *T. nanus* in each period [71] (Figure 11 and Figure 12, Appendix A). *T. nanus*’s centroid is located in the central area of Chuxiong in all climate scenarios. Since the LIG period, the centroid of *T. nanus* has migrated to northeast, as a whole. At present, it is distributed in the most northern range. The distance of the centroid migration from the LIG to LGM is the longest, which is 17.71 km, and the distance of the centroid migration in the MH is the shortest, which is only 6.86 km. Under the SSP1-2.6 scenario, the centroid will generally move to southern and high-altitude areas in the future. From the present to 2030, the migration distance of the centroid is the longest, which is 13.28 km, and from 2030 to 2050, the migration distance is the lowest, which is 4.71 km, and then the migration distance increases slowly. Under the SSP2-4.5 scenario, the centroid will generally migrate southwest. From 2030 to 2050, the migration distance of the centroid is the longest, which is 16.10 km, and the altitude is the lowest in this period. From 2030 to 2050, the migration distance is the lowest, which is 4.71 km, and then gradually decreases. Under the SSP5-8.5 scenario, the centroid will generally move south. From 2070 to 2090, the change in the distance of the centroid is the longest, which is 16.67 km. And from 2030 to 2050, the migration distance is the shortest, which is 3.14 km.

## 4. Discussion

### 4.1. Model Rationality and Prospects

*T. nanus* is one of the 231 target species in the “Investigation and germplasm resources conservation program of wild plants with extremely small populations in Southwest China” [39]. Currently, due to the influence of human activities and habitat fragmentation, the rare and endangered minimal population has low genetic diversity, poor adaptability, and a narrow distribution, which may lead to species extinction [72,73]. Therefore, it is of great significance that we simulate the suitable habitat of rare and endangered species based on a niche model [74,75]. For example, *Ostrya rehderiana* is a famous plant species with extremely small populations. The prediction results based on the MaxEnt model show that the accuracy is high, and its suitable area will increase in the future [76]. Precipitation and temperature are the factors limiting the distribution of *Firmiana kwangsiensis*, and its distribution area will increase in the future [77]. In the study of *Ulmus elongate*, Zhang et al. shows that its suitable area will increase in the future, and there is a trend of migration to a high altitude [78]. In this study, the results are similar to those of *Ostrya rehderiana*, *Firmiana kwangsiensis*, and *Ulmus elongate*. The simulated potentially suitable area for *T. nanus* is consistent with its actual distribution area, which further shows that the MaxEnt model has good predictability for use regarding rare and endangered populations.

However, in the current SDMs, the results predicted by a niche model usually represent a fundamental niche, not a realized niche [79]. This is because certain factors, (1) such as the competition between organisms and the relationship between animals and plants, are not taken into account when predicting the optimal distribution of a species [80], e.g., in the study of avains, its distribution is most related to food conditions, rather than various climatic factors [81]; (2) the model does not consider stochastic processes, such as dispersal limitation, habitat connectivity, and the health status of destination habitats, which will lead to the realized niche of species being much smaller than the predicted distribution [82,83]. Therefore, in the future, we urgently need to adopt multidisciplinary methods, such as integrating species distribution models and constructing new models (covering abiotic and biotic factors) to better understand the adaptation and the mechanisms behind the survival of the studied species.

### 4.2. Main Environmental Factors Affecting the Distribution of T. nanus

The most important environmental factors affecting *T. nanus* are the annual precipitation (bio12), mean temperature of the coldest quarter (bio11), temperature seasonality (bio4), the precipitation of the wettest quarter (bio16), and the isothermality (bio3). *Trachycarpus* plants most likely originate from Laurasia and Western Gondwana, where there were tropical and subtropical climates [84]. With the continuous increase of heat and water, *Trachycarpus* species will become gradually dense [85]. In addition, drought stress can reduce the photosynthesis and respiration rate of *Trachycarpus* plants, and low temperature stress can damage the permeability of its cell membrane [86,87]. Bioclimatic factors can directly affect the growth and development of plants and seedling regeneration [88], but these are performed differently among species. For example, *Euscaphis japonica* mainly responds to temperature [24], but *Quercus glauca* responds to annual precipitation [57]. In this study, the influence of precipitation-related factors on *T. nanus* is greater than that of temperature-related factors. The central Yunnan Plateau, where *T. nanus* is located, is a region with a seasonal drought [89]. Arid conditions are more favorable than cold temperatures to control the plant’s distribution [90]. Drought will destroy the structure and function of cells, resulting in the reduction of photosynthesis, seriously affecting the growth and development of plants [91]. *T. nanus*’ seeds are sensitive to dehydration and highly susceptible to water stress [1].

The change in environmental factors will not only have a great impact on an endangered species, but also enlarge the loss of biodiversity. Because the survival of plants is not isolated, but in a certain interdependent relationship with other species [92], the disappearance of a species often leads to the coexistence crisis of 10–30 species [93]. Therefore, the changes in temperature and precipitation caused by climate change have a far greater impact on an endangered species itself, and may change the structure and function of the ecosystem [94]. Global warming usually leads to the advancement of plant phenology [95]. However, the same period of precipitation and heat in the distribution area of *T. nanus* are important factors for its growth [33], and the increase in temperature will increase the occurrences of drought [96], which will most likely cause the growth of *T. nanus* to be limited by rainfall. The growth of *T. nanus* is slow and its regeneration ability is relatively weak [33]. When its own habitat is suppressed, its ability to expand and compete for a niche is weakened. Meanwhile, climate changes may lead to ecosystems’ degradation by means such as flash floods, droughts, and forest fires, which further aggravates the extinction possibility of endangered plants.

### 4.3. Spatial Distribution Change in Potentially Suitable Areas for T. nanus

Our predicted results are similar to a former field survey [28]. The results show that the simulation accuracy of the model is high. Under the past and three future climate scenarios of SSP1-2.6, SSP2-4.5, and SSP5-8.5, the habitat suitability of *T. nanus* changes differently. From the LIG to the MH, the potential suitable area first increases and then decreases, but the change rate decreases firstly and then increases. In LGM, the potential habitat area reaches the maximum and the rate of change is the minimum. However, the overall trend is gradually increasing (Appendix A). Sporopollen records during the LGM show that although global cooling and precipitation decreased generally, it had little impact on the subtropical region [97]. At the same time, the geographical position and mountainous landform of Yunnan Province leads to the stability of its climatic conditions, and the occurrence of the Quaternary Ice Age has less influence on the plants in Yunnan [98]. During the LIG period and the MH, the temperature was high, but the precipitation was low and the sunlight was strong, and the temperature changed sharply [97]. *T. nanus* is a shade-tolerant species, and strong sunlight and less precipitation can easily cause seed dehydration and affect seed dispersal [39]. At present, the potentially suitable area for *T. nanus* is larger than in the past, but its suitable area is relatively scattered and seriously fragmented, which is speculated to be related to human interference and land use change [99]. A very convincing example here is what Gibbons and Spanner described when they first visited the endangered *T. nanus* in 1993, “With the exception of the steepest slopes, all of the land was under cultivation, much of it grazed by cattle and goats, which, although undoubtedly finding the adult palm leaves too tough to handle, probably found the inflorescences and young seedlings tasty and edible. This would account for the few seeds and total absence of young and juvenile plants” [29].

Under the SSP2-4.5 scenario, the potentially suitable area and the change rate of *T. nanus* will decrease with time. We believe that this is due to the currently unbalanced global development, which is closest to our current socio-economic development characteristics [100]. The temperature will rise by about 2 °C. And the global precipitation will also increase, but the precipitation to the south of the Yangtze River, especially in Yunnan, will drop by about 15% [101]. It is speculated that this is the reason for the decline in the area of *T. nanus* during this period. Under the scenarios of SSP1-2.6 and SSP5-8.5, the potentially suitable area and the change rate of *T. nanus* showed a gradual growth trend, but SSP5-8.5 increased rapidly compared with SSP1-2.6. We think it may be related to the sustainable development path of SSP1-2.6, and the slow growth in temperature and precipitation. While there is a large increase in temperature and precipitation in SSP5-8.5 [102], under ideal conditions, it may increase the niche space of *T. nanus* and expand its suitable growth area [103]. However, irregular temperature and precipitation, frequent extreme events [104], and its difficulties in regeneration and long-distance dispersal may actually accelerate its extinction.

The change in the suitable areas for *T. nanus* may also have a certain impact on the ecosystem. First of all, although this species exists as a companion species in the community, other plant and animal species must have a certain dependence on its existence [92]. For example, *T. nanus* provides food for some birds and ferrets in the local ecosystem (field observation), and the change of its suitable area may lead to the destruction of a food chain or ecological network. Secondly, *T. nanus* extinction caused by habitat shifts will reduce the species richness of ecosystems. Finally, the change in the suitable area of endangered species may affect the local community’s composition and unique ecosystem functions.

Therefore, according to the characteristics of different climate scenarios (SSP1-2.6, SSP2-4.5, and SSP5-8.5) [104], we put forward different management strategies for planning the protection of *T. nanus*. The SSP1-2.6 scenario has a low-carbon emission and highly sustainable development. Under this scenario, climate change is effectively controlled and there may be a moderate trend of warmth and humidity. This has a relatively positive impact on the living environment of *T. nanus*. Conservation planning and management strategies should focus on maintaining the suitable area of *T. nanus*. The SSP2-4.5 scenario has a moderate carbon emission and a moderate development path. Under this scenario, the temperature will rise and precipitation will change, resulting in a certain impact on *T. nanus*. Protection strategies need to actively respond to the climate change impact, such as strengthening the continuous monitoring of *T. nanus* growth and the management of protected areas. Scenario SSP5-8.5 is a development path with high carbon emissions and high inequality. Under this scenario, greenhouse gas emissions are high and there are many extreme events, which may lead to serious climate change and environmental damage. This may pose a serious threat to *T. nanus*. Protection enforcement, and its emergent implementation via ecological restoration are necessary for this situation. Multiple measurements should be applicable, such as in situ and ex situ conservation, genetic resources’ recovery, intense monitoring, etc. The planning and management strategies of *T. nanus*’ protection need to be adjusted and customized according to climate scenarios. It is important to take effective measures in time to ensure the long-term survival and health of *T. nanus* populations and their habitats.

### 4.4. Centroid Analysis for T. nanus

According to the migration direction of the centroid (Figure 11 and Appendix A), *T. nanus* migrates to the northeast greatly from the LIG to the LGM, and then the centroid tends to be stable from the LGM to the current day. In accordance with the latitude change, its distribution altitude is the highest in LIG (2541 m), and is the lowest in the LGM (2044 m). It is speculated that the temperature of the LIG was about 2 °C higher than the current temperature, while the sudden drop in temperature in the LGM led to the migration of plants to a lower altitude to seek “shelter” and greatly reducing their potentially suitable area [105,106]. In MH, although the temperature was lower than that of the LIG period, it was indeed a warming period, which promoted the expansion of plants’ suitable habitats and their migration toward high altitudes [107,108]. In the future, with the increases in temperature and precipitation, the potential habitat of *T. nanus* will generally expand to a higher altitude to enlarge its niche, and the areas that may be unsuitable for species establishment before will have become more favorable [109]. These phenomena are found in California endemic flora [110], *Castanopsis carlesii* [111], and *Rana spinosa* [65], etc. The distribution of *T. nanus* is the lowest at elevations above sea level in the SSP5-8.5 scenario in 2090. We speculate that the temperature rise might be the fiercest in 2090, but the irregular precipitation will lead to the enhancement of water availability and evaporation, which leads to the migration of plants to a lower altitude.

Convincing evidence shows that climate change will be one of the important driving factors for species extinction [8,112]. In the future scenario, the centroid of *T. nanus* will move a long distance. The colonization of plant species is a complex process, which is determined by fertility, seed dispersal potential, and suitable habitat availability [113]. It is predicted that the speed of climate change in the future will be much faster than that after the glacial age, and plants will need higher mobility [114]. However, it is worried that (1) many species find it more and more difficult to migrate due to human-driven habitat fragmentation [115,116], and (2) the acceleration of global warming makes it necessary for species to spread faster [117]. Therefore, *T. nanus* may not keep up with the rapidly changing climate. We found that the centroid of *T. nanus* has moved about 35 km in the past tens of thousands of years, and it has to approach or exceed this value in less than a hundred years in the future (Figure 12). The dispersal ability of species is limited in this century, usually within 10–20 km per century of expanding the current border [118]. The distributions of species have recently shifted to a higher elevation at a median rate of 11.0 m per decade, and to a higher latitude at a median rate of 16.9 km per decade [9]. Therefore, the rapid process of climate change may make many species unable to evolve the necessary adaptabilities [119]. As shown by ecological theory and conservation biology, the modern landscape provides little flexibility for ecosystems to adapt to rapid environmental changes. In contrast to the geo-historical migration process, species in many areas now have to cross the increasingly impassable landscapes caused by intensified human activities [120]. Due to the extensive loss and fragmentation of habitats, many areas that may be suitable are far from the current distribution and beyond the dispersal capacity of many species. Therefore, species with low adaptability or vagility will fall into the dilemma of forced climate change and have a low possibility of finding distant habitats to settle down, which will eventually lead to an increase in their extinction rate [7]. Species that are expected to expand or change the range of their suitable conditions may not realize their potential new range due to dispersal limitation. Therefore, we have reason to think that continued climate change might lead to a “nowhere to run” situation for *T. nanus*. This study infers that *T. nanus* is threatened by climate change, and it may be more endangered in the future.

### 4.5. Protection Countermeasures and Suggestions for T. nanus

It was found that the current threats to the endangered *T. nanus* were mainly caused by human impacts [28], e.g., wind-power development, medicinal use, grazing, deforestation, etc. The habitat fragmentation caused by human activities reduces the connectivity between populations, as well as plant–animal interaction and seed dispersal [121]. Plants adapt to climate change based on their biological properties [122]. *T. nanus* has a long life span and slow regeneration ability; it takes about 10 years, that from seed germination to adult individuals, to begin flowering and fruiting [33]. Therefore, it is difficult for *T. nanus* to expand its actual niche under rapid climate change. The growth of *T. nanus* seedlings requires a shadow habitat. Commonly, it is a companion species in pinewood or ever-green board-leaved forests [32], and with the reduction in forest areas [123], the regeneration of *T. nanus* seedlings would be influenced.

Establishing nature reserves is an important protection strategy. For example, the in situ and ex situ conservation of the extremely small population of *Salvia daiguii* saved this species [124]. The endangered species *Saraca asoca* has greatly improved its growth in a nature reserve [125]. The Yellowstone–Yukon Ecological Corridor promotes the migration of various animals and plants and protects the habitats of many endangered species, including gray wolves and bears [126]. Nature reserves and ecological corridors provide good habitats and interconnected channels for organisms in different regions, which enhance biodiversity and ecosystem health. At the same time, they also promote the animal dispersal of *T. nanus*’ seeds to enhance gene flow and population regeneration. Established nature reserves only cover a small part of the suitable areas for *T. nanus* (Appendix A). Because most *T. nanus* populations have not been included in the scope of protection [41], we strongly suggest that more small, protected sites be established, as well as corridors for population connectivity to ensure the proliferation and recovery of *T. nanus* under climate change [127].

In addition, local people may have rich knowledge of the distribution and growth habits of this species, and the forestry department (in charge of endangered plants’ protection) needs to cooperate with the local communities to jointly conduct their protection plan. Strengthening local people’s awareness on *T. nanus*’ protection will reduce anthropogenic threats, such as picking flowers and fruits and stock grazing. The implement of *T. nanus*’ protection will facilitate afforestation and improve habitat quality, which benefits local communities. Sustainable *T. nanus* protection should be enhanced by reasonable resource exploration, and more studies are needed. *T. nanus* has a potential value in gardening, as appropriate artificial plantation and breeding will promote wild population protection. Researchers should carry out research on germplasm resources and promote genetic improvement. The state and local governments and local stakeholders should work together to facilitate its protection in a sustainable way.

## 5. Conclusions

In this study, the MaxEnt model based on optimization has achieved satisfactory results. Bio12, bio11, bio4, bio16, and bio3 are the most important environmental factors affecting *T. nanus* distribution, among which precipitation is more prominent. Since the LIG period, the potentially suitable area for *T. nanus* is slightly lower today and the change rate of suitable areas is slow. In the future, the potentially suitable area for *T. nanus* will change faster. This may pose a great threat to the viability of the population and may increase the extinction risk of *T. nanus*. Less climate change will reduce the habitat shifts of *T. nanus* and the direct influences on its growth. The positive conservation of *T. nanus* relies on global carbon neutrality efforts. Global carbon neutrality needs the joint efforts and policy support of the international community to reduce greenhouse gas emissions. At present, due to long-term human disturbances, most of *T. nanus*’s habitat has not been effectively protected. Facing climate change threats, it is important to establish more protection sites and corridors to mediate its habitat’s shift. In addition, local community engagement in conservation should be encouraged, such as joint management, mass education, sustainable resource exploration, etc. This study enriches the conservation biology of endangered plant species, and can provide an important reference for conservation strategies.

## Figures and Tables

**Figure 1 biology-13-00240-f001:**
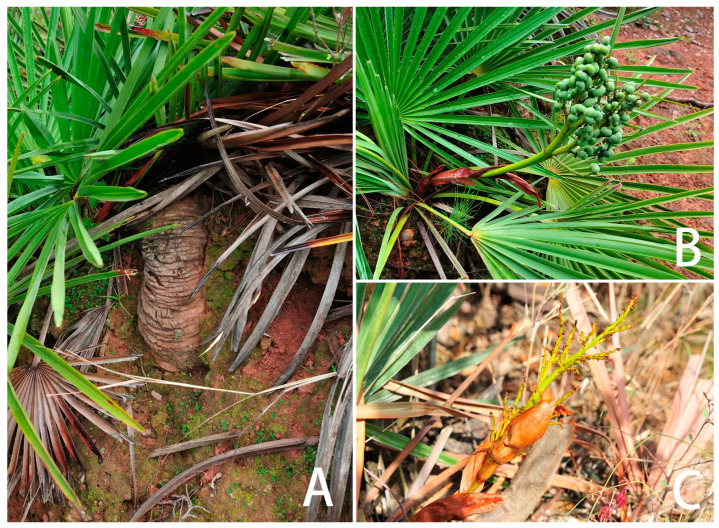
Morphological characteristics of *T. nanus* in Yunnan, China. (**A**) Underground stem; (**B**) female plant; (**C**) male plant. Photograph: Chongyun Wang.

**Figure 2 biology-13-00240-f002:**
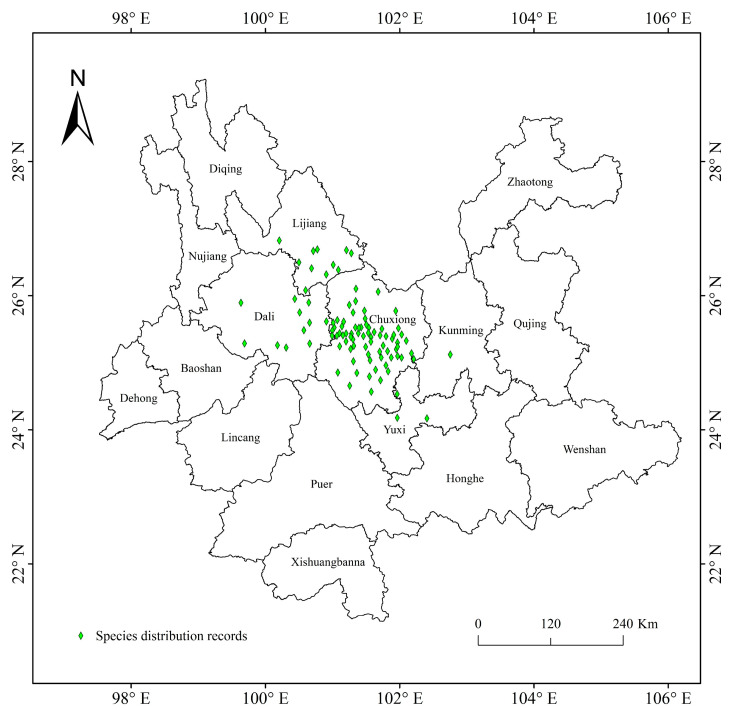
Geographical distribution records of *T. nanus* in Yunnan, China.

**Figure 3 biology-13-00240-f003:**
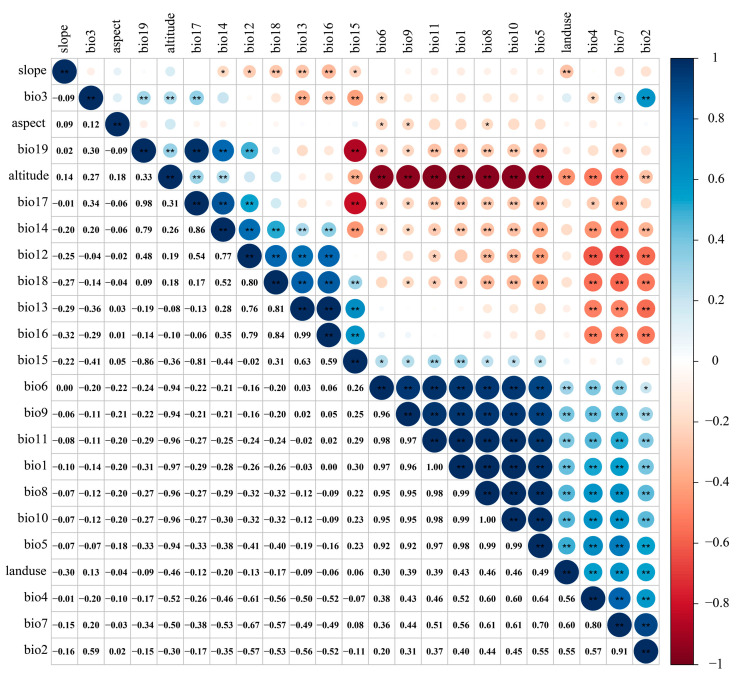
Environmental factor correlation heat map.“**” means *p* < 0.01, “*” means *p* < 0.05.

**Figure 4 biology-13-00240-f004:**
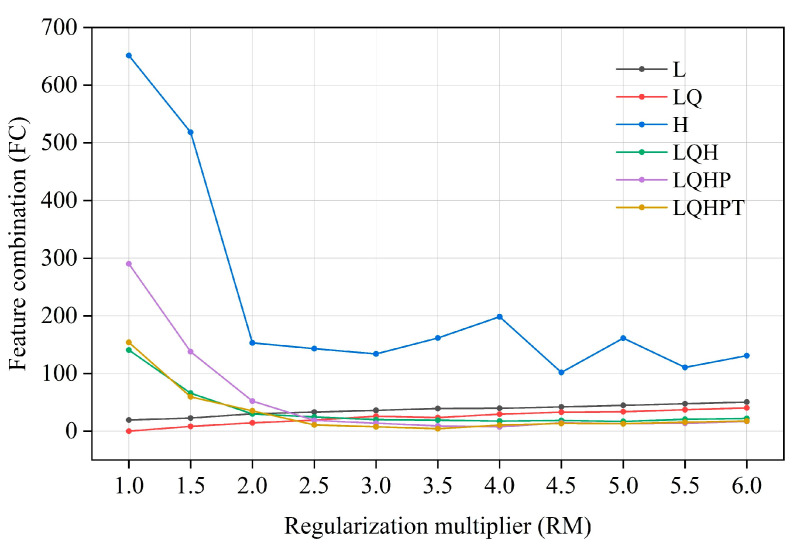
Evaluation results of the MaxEnt model under different settings.

**Figure 5 biology-13-00240-f005:**
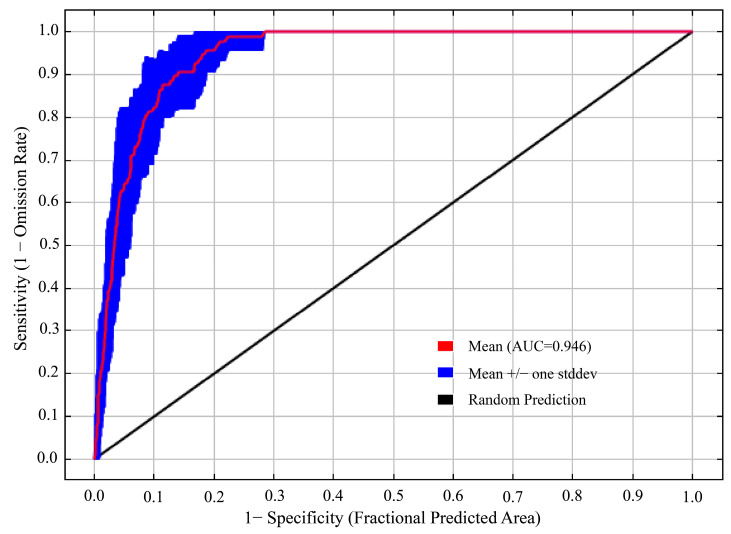
AUC value of *T. nanus* predicted by the MaxEnt model.

**Figure 6 biology-13-00240-f006:**
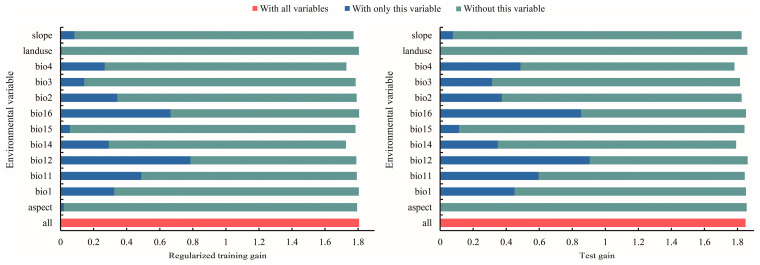
Evaluation of major environmental factors using the jackknife method.

**Figure 7 biology-13-00240-f007:**
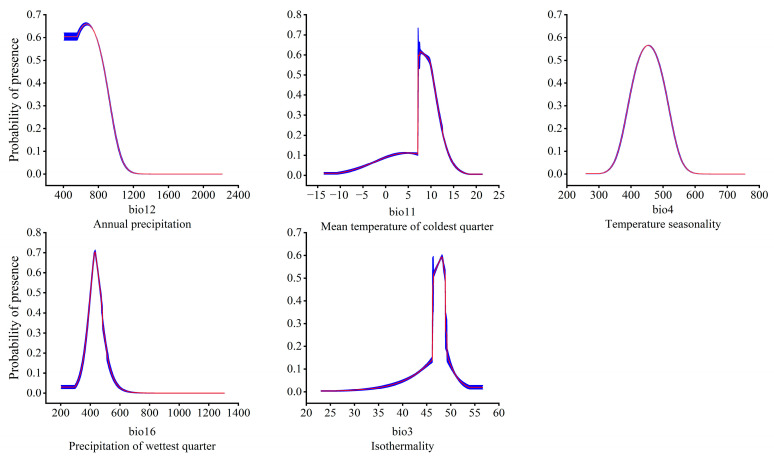
Relationship between the potentially suitable area and the single factor response curve. The red curves represent average value over 10 replicate runs, while blue margins represented ± SD calculated for 10 replicates.

**Figure 8 biology-13-00240-f008:**
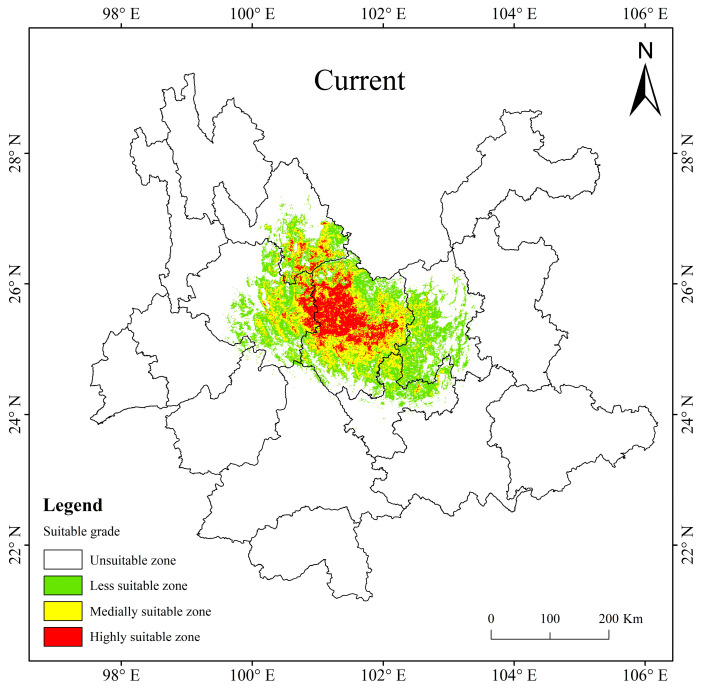
The potentially suitable areas for *T. nanus* in Yunnan Province under current climate conditions.

**Figure 9 biology-13-00240-f009:**
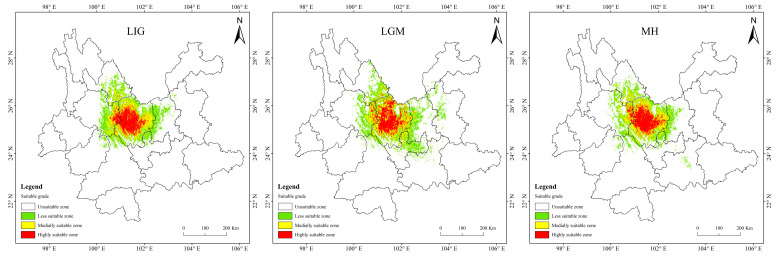
Prediction of potentially suitable areas for *T. nanus* during different periods.

**Figure 10 biology-13-00240-f010:**
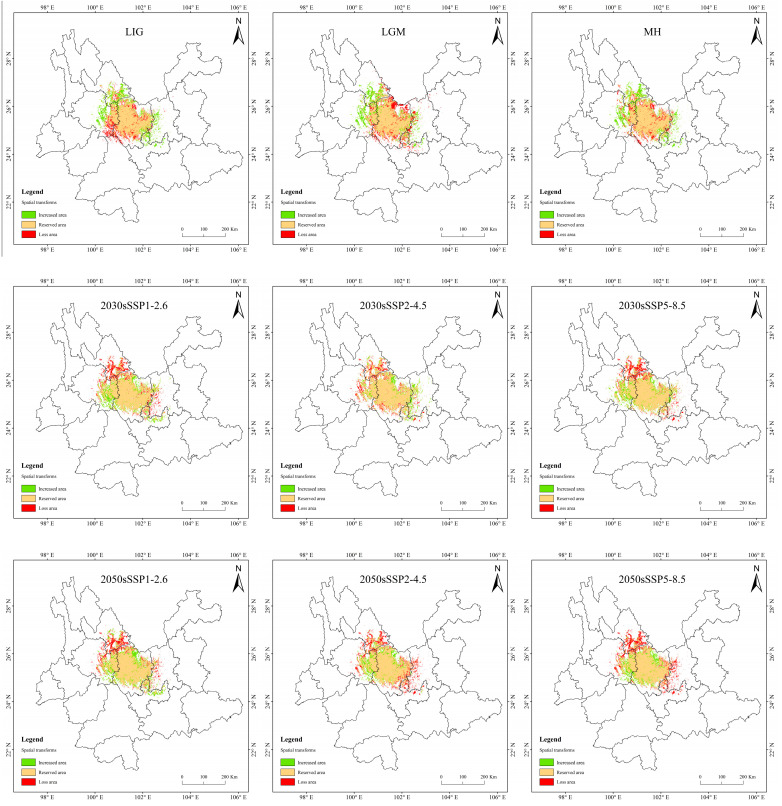
Suitable distribution changes of *T. nanus* under different climate change scenarios.

**Figure 11 biology-13-00240-f011:**
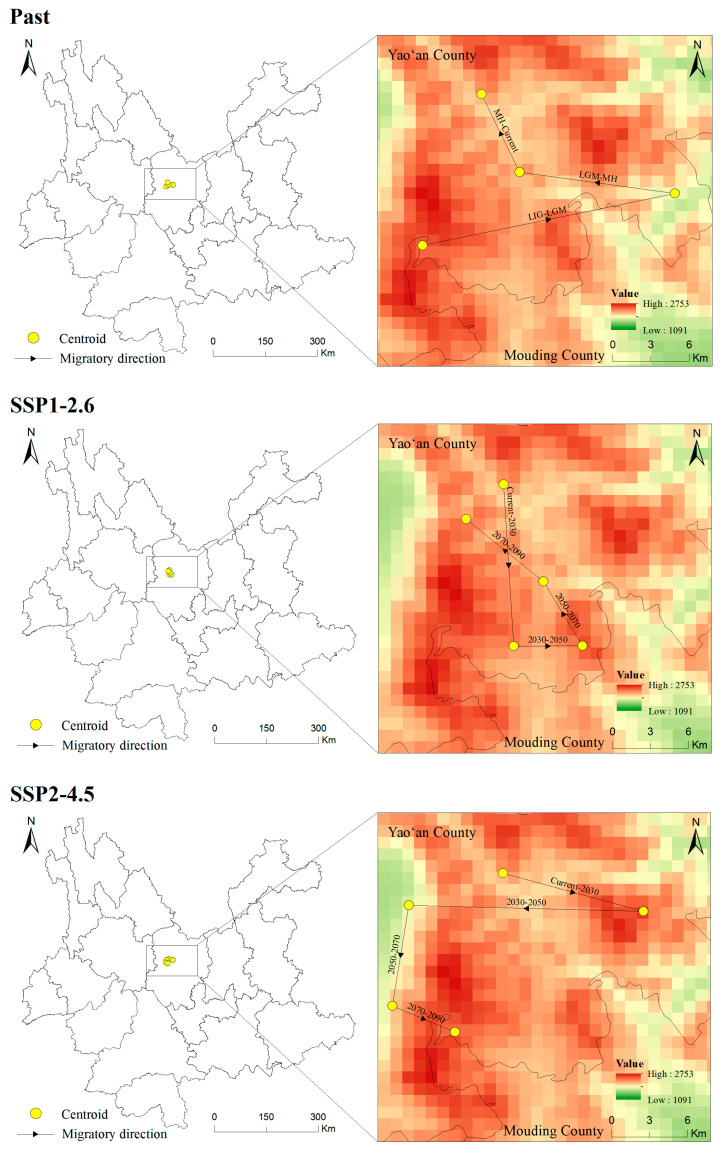
Location of centroid shifts of suitable areas for *T. nanus* under different climate scenarios in different periods.

**Figure 12 biology-13-00240-f012:**
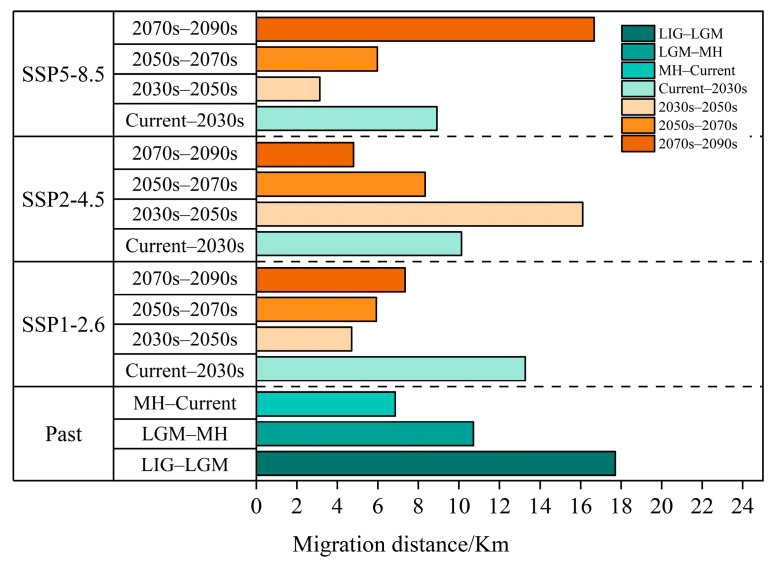
Migration distance of the centroids in the past and under different scenarios in the future.

**Table 1 biology-13-00240-t001:** The importance of environmental variables on the distribution of *T. nanus*.

Variable	Description	Unit	Percent Contribution/%	Permutation Importance/%
bio12	Annual precipitation	mm	55.1	34.4
bio4	Temperature seasonality	-	11.7	15.0
bio11	Mean temperature of coldest quarter	°C	7.6	0.2
bio15	Precipitation seasonality	-	5.2	0.5
bio14	Precipitation of driest month	mm	3.6	3.8
Alt	Elevation	m	2.7	13.1
asp	Aspect	°	2.5	1.1
Slo	Slope	°	2.3	0.9
Landuse	Land use type	-	1.6	0.6
bio3	Isothermality	-	1.2	7.4
bio1	Annual mean temperature	°C	1.1	12.9
bio16	Precipitation of wettest quarter	mm	1.1	1.3
bio9	Mean temperature of driest quarter	°C	0.8	2.8
bio7	Temperature annual range	°C	0.7	2.8
bio6	Min temperature of coldest month	°C	0.7	2.5
bio13	Precipitation of wettest month	mm	0.6	0.1
bio18	Precipitation of warmest quarter	mm	0.3	0.3
bio17	Precipitation of driest quarter	mm	0.3	0.0
bio2	Mean diurnal range	°C	0.2	0.0
bio8	Mean temperature of wettest quarter	°C	0.2	0.1
bio19	Precipitation of coldest quarter	mm	0.1	0.1
bio10	Mean temperature of warmest quarter	°C	0.1	0.0
bio5	Max temperature of warmest month	°C	0.1	0.5

**Table 2 biology-13-00240-t002:** Various parameters of the main environmental variables of *T. nanus*.

Variable	Description	Unit	Percent Contribution/%	Permutation Importance/%
bio12	Annual precipitation	mm	48.9	57.2
bio11	Mean temperature of coldest quarter	°C	11.3	1.3
bio4	Temperature seasonality	-	9.6	24.8
bio16	Precipitation of wettest quarter	mm	8.0	0.5
bio3	Isothermality	-	7.1	5.4
bio2	Mean diurnal range	°C	5.3	2.5
bio15	Precipitation seasonality	-	4.9	1.9
bio1	Annual mean temperature	°C	1.1	0.4
asp	Aspect	°	1.1	0.5
bio14	Precipitation of driest month	mm	0.9	4.5
slope	Slope	°	0.9	0.6
Land use	Land use type	-	0.8	0.3

**Table 3 biology-13-00240-t003:** Evaluation metrics of the MaxEnt model generated by ENMeval.

Type	RM	FC	Delta. AICc	Avg. Diff. AUC	Mean. OR_10_
Default	1	LQHPT	153.875	0.038	0.292
Optimized	1	LQ	0	0.031	0.115

FC: feature combination; RM: regulatory multiplier; delta. AICc: the minimum information criterion AICc value; Avg. diff. AUC: difference between the AUC value; Mean. OR_10_: mean value of the 10% training omission rate.

**Table 4 biology-13-00240-t004:** Suitable areas for *T. nanus* in Yunnan Province at different times and under different climate scenarios (×10^4^ km^2^).

Period	Less Suitable Area	Medially Suitable Area	Highly Suitable Area	Total Suitable Area
LIG	2.43	1.43	1.09	4.95
LGM	2.87	1.59	1.15	5.61
MH	2.16	1.37	1.11	4.64
Current	2.86	1.79	1.00	5.65
2030s-SSP1-2.6	2.96	1.83	1.12	5.91
2030s-SSP2-4.5	2.46	1.57	1.07	5.10
2030s-SSP5-8.5	2.67	1.58	1.04	5.29
2050s-SSP1-2.6	2.56	1.76	1.23	5.45
2050s-SSP2-4.5	2.96	1.82	1.01	5.79
2050s-SSP5-8.5	2.75	1.71	1.15	5.61
2070s-SSP1-2.6	2.77	1.92	1.12	5.81
2070s-SSP2-4.5	2.38	1.43	1.06	4.87
2070s-SSP5-8.5	3.12	2.16	1.21	6.49
2090s-SSP1-2.6	2.61	1.74	1.13	5.48
2090s-SSP2-4.5	2.48	1.54	1.13	5.15
2090s-SSP5-8.5	3.29	2.63	1.16	7.08

**Table 5 biology-13-00240-t005:** Suitable distribution changes of *T. nanus* under different climate change scenarios.

Period	Area/×10^4^ km^2^				Change Rate/%			
	Increase	Reserved	Lost	Change	Increase	Reserved	Lost	Change
LIG	0.81	1.99	0.54	0.27	14.34	35.22	9.56	4.78
LGM	0.79	2.03	0.72	0.07	13.98	35.93	12.74	1.24
MH	0.77	2.03	0.45	0.32	13.63	35.93	7.96	5.66
2030s-SSP1-2.6	0.63	2.26	0.54	0.09	11.15	40.00	9.56	1.59
2030s-SSP2-4.5	0.50	2.34	0.46	0.04	8.85	41.42	8.14	0.71
2030s-SSP5-8.5	0.58	2.28	0.51	0.07	10.26	40.35	9.03	1.24
2050s-SSP1-2.6	0.68	2.27	0.53	0.15	12.04	40.18	9.38	2.66
2050s-SSP2-4.5	0.47	2.18	0.61	−0.14	8.32	38.58	10.80	−2.48
2050s-SSP5-8.5	0.49	2.12	0.67	−0.18	8.67	37.52	11.86	−3.19
2070s-SSP1-2.6	0.70	2.34	0.45	0.25	12.39	41.42	7.96	4.42
2070s-SSP2-4.5	0.42	2.06	0.73	−0.31	7.43	36.46	12.92	−5.49
2070s-SSP5-8.5	1.02	2.36	0.44	0.58	18.05	41.77	7.79	10.27
2090s-SSP1-2.6	0.61	2.27	0.53	0.08	10.80	40.18	9.38	1.42
2090s-SSP2-4.5	0.59	2.07	0.73	−0.14	10.44	36.64	12.92	−2.48
2090s-SSP5-8.5	1.32	2.47	0.32	1.00	23.36	43.72	5.66	17.7

## Data Availability

All data generated by this study are available from the corresponding author upon reasonable request.

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
