# Peer review of "Response of Extremely Small Populations to Climate Change—A Case of Trachycarpus nanus in Yunnan, China"

_biology, 2024, doi:10.3390/biology13040240_

Round 1

Reviewer 1 Report

Comments and Suggestions for Authors

Climate change is among the important issues that need to be addressed in recent years. Because this change will affect access to food in the future. I congratulate the authors for doing such a work. The article is generally well prepared, the sources used are sufficient, and it is well discussed with other studies. It will make important contributions to science. But I have some small suggestions;

The summary is too long, please shorten it.

It would be more appropriate to use square brackets when using references. For example, it is confusing to cite the source without square brackets in the introduction section.

The resolution of the figures used in the study is low, it is recommended to increase them.

Throughout the study, there were missed punctuation marks and these should be corrected.

Additionally, there are errors in the spelling of abbreviations and symbols in some places.

Some sentences may seem repetitive.

The conclusion section should be rewritten in a meaningful way.

These corrections can be published after they are made.

Author Response

Comments and Suggestions for Authors:

Climate change is among the important issues that need to be addressed in recent years. Because this change will affect access to food in the future. I congratulate the authors for doing such a work. The article is generally well prepared, the sources used are sufficient, and it is well discussed with other studies. It will make important contributions to science. But I have some small suggestions;

Comments 1: The summary is too long, please shorten it.

Response 1: Thanks for your comments. The summary and abstract have been shortened as required by biology.

Comments 2: It would be more appropriate to use square brackets when using references. For example, it is confusing to cite the source without square brackets in the introduction section.

Response 2: Thanks for your suggestion. We have changed them all as required.

Comments 3: The resolution of the figures used in the study is low, it is recommended to increase them.

Response 3: The resolution of the figures has been improved to illustrate the results.

Comments 4: Throughout the study, there were missed punctuation marks and these should be corrected.

Response 4: Punctuation errors have been carefully checked, and corrected in full text.

Comments 5: Additionally, there are errors in the spelling of abbreviations and symbols in some places.

Response 5: Thanks for your comments. The spelling errors of abbreviations and symbols have been corrected.

Comments 6: Some sentences may seem repetitive.

Response 6: Thanks for your comments. After careful examination, redundant sentences have been deleted.

Comments 7: The conclusion section should be rewritten in a meaningful way.

Response 7: Thank you very much for your suggestion. The conclusion has been rewritten in a meaningful way. The revised conclusion is in Line 636-652 in manuscript-clean.

Reviewer 2 Report

Comments and Suggestions for Authors

This paper presents an insightful and highly relevant investigation into the effects of climate change on the distribution of Trachycarpus nanus, a rare and endangered species native to Yunnan Province, China. By employing the MaxEnt model optimized for this specific study, the research provides a comprehensive analysis of the current and future distribution patterns of T. nanus, grounded in both historical and projected climate scenarios.

Introduction

  1. While the introduction effectively sets up the context and methodology, it could be strengthened by more explicitly stating the gap in the current literature that your study aims to fill. This might include the novel application of SDMs to T. nanus or addressing specific knowledge gaps in the conservation strategies for this species.
  2. While human impacts are mentioned as one of the factors influencing plant distributions, the introduction could benefit from a brief discussion on how specific human activities (e.g., deforestation, urbanization) might be impacting T. nanus.
  3. The introduction uses several technical terms and abbreviations (e.g., SDM, LIG, LGM, MH). Ensuring that all such terms are clearly defined when first introduced will make the paper more accessible to a broader audience, including non-specialists.

Method

  1. While you mention filtering distribution records to avoid spatial autocorrelation, expanding on how you determined which points to retain or remove could provide more insight into the potential biases or limitations of your dataset.
  2. Meanwhile, based on the preliminary simulation results of MaxEnt model (Table 1), environmental factors with correlation ≥0.8 and low contribution rate were eliminated. I think the threshold is higher it should be less.
  3. Although MaxEnt is a well-known model for species distribution modeling, briefly discussing why it was chosen over other models, based on your specific research questions and data characteristics.
  4. Mentioning the spatial resolution is good, but discussing the implications of this choice on the accuracy and granularity of your predictions would be insightful. For instance, how does the 1km resolution affect the modeling of a species with small, scattered populations?
  5. For readers less familiar with the specific models and statistical methods used, a brief explanation of terms like "regularized multipliers" and "feature types" in the context of MaxEnt modeling could improve the accessibility of your methodology.

Results

  1. While the results are robust, further clarifying how environmental factors were weighted or integrated into the model could provide deeper insights into their relative importance and interactions.
  2. Expanding on the ecological implications of the dominant environmental factors and habitat shifts could enrich the discussion. For instance, how do these shifts correlate with broader ecological trends in Yunnan or similar regions?
  3. While the paper mentions different climate scenarios, a more detailed discussion on how each scenario could specifically impact conservation planning and management strategies for T. nanus would be beneficial.

Discussion

  1. While the discussion on the main environmental factors and model predictions is robust, expanding on the broader ecological implications of these changes could provide a more comprehensive view.
  2. The call for more protected areas and corridors is well-justified, but providing more specific recommendations or potential models for these conservation strategies could enhance the section. This might include examples of successful conservation efforts for similar species or habitat types, which could serve as a blueprint for T. nanus.
  3. Discussing potential adaptation and mitigation strategies for the impacts of climate change on T. nanus would be a valuable addition.
  4. While the discussion acknowledges human impacts on T. nanus, elaborating on strategies for engaging local communities in conservation efforts could add depth. Community-based conservation strategies could be particularly relevant for species affected by habitat fragmentation and land-use changes.

Conclusion

  1. While you suggest reducing human interference and protecting indigenous plant communities, elaborating on specific measures (e.g., sustainable land-use practices, and community engagement in conservation) could provide a more actionable framework for stakeholders.
  2. Addressing the uncertainties in climate change projections and their implications for T. nanus could lead to discussions on adaptive management strategies.
  3. Expanding on how global carbon neutrality efforts could mitigate climate change impacts on T. nanus would provide a hopeful perspective on the potential for policy actions to influence conservation outcomes positively.
Comments on the Quality of English Language

Minor editing of English language required

Author Response

Response to Reviewer 2’s Comments

Comments and Suggestions for Authors

This paper presents an insightful and highly relevant investigation into the effects of climate change on the distribution of Trachycarpus nanus, a rare and endangered species native to Yunnan Province, China. By employing the MaxEnt model optimized for this specific study, the research provides a comprehensive analysis of the current and future distribution patterns of T. nanus, grounded in both historical and projected climate scenarios.

Introduction

Comments 1: While the introduction effectively sets up the context and methodology, it could be strengthened by more explicitly stating the gap in the current literature that your study aims to fill. This might include the novel application of SDMs to T. nanus or addressing specific knowledge gaps in the conservation strategies for this species.

Response 1: Thanks for your suggestion. In fact, it is first report on this unique palm shrub’s present habitat and threats under climatic change by MaxEnt modeling. And we used optimization methods to improve modeling results. Our results first comprehensively address its present distribution and potential habitat factors, and will be helpful for its conservation strategies. Introduction is improved as suggested.

In our manuscript, this part is in the first paragraph on page 3, lines 102-109 in manuscript-clean.

Comments 2: While human impacts are mentioned as one of the factors influencing plant distributions, the introduction could benefit from a brief discussion on how specific human activities (e.g., deforestation, urbanization) might be impacting T. nanus.

Response 2: Thanks for your suggestion. Based on former researches and field observation, human activities such as deforestation for agricultural practice, leaf-cutting for brooms, goat grazing, road and wind power farm construction. Brief discussion is improved in relative context.

In our manuscript, this part is in the third paragraph on page 2, lines 83-93 in manuscript-clean. The specific contents are as follows:

“The development zones with good wind energy resources in Yunnan Province cover most areas where the T. nanus is distributed [34]. Wind power farm construction put threats on its habitats [35–36]. In addition, livestock overgrazing (Figure S1–B, C) [29] and road construction [28] are also human activities that threaten T. nanus. According to the field investigation and literature, the invasive species, Ageratina adenophora has occupied a large number of T. nanus’ habitats (Figure S1–D), seriously affecting the survival and regeneration of its seedlings [37]”.

Comments 3: The introduction uses several technical terms and abbreviations (e.g., SDM, LIG, LGM, MH). Ensuring that all such terms are clearly defined when first introduced will make the paper more accessible to a broader audience, including non-specialists.

Response 3: Thanks for your suggestion. The technical terms and abbreviations used in the introduction (such as SDM, LIG, LGM, MH), we checked again and ensured that all these terms are clearly defined when we first introduce them.

Method

Comments 1: While you mention filtering distribution records to avoid spatial autocorrelation, expanding on how you determined which points to retain or remove could provide more insight into the potential biases or limitations of your dataset.

Response 1: Thanks for the suggestion. We made a statement in line 129-138 in manuscript-clean, relative text is improved as follows:

“Firstly, by observing the spatial distribution of the T. nanus distribution points, it is found that some recording points are in an aggregation state and there is a problem of spatial autocorrelation, which may be caused by sampling deviation or the limitation of data collection methods. When there are local spatial clusters, the modelling is usually over-fitted (reducing the prediction ability of the model) and exaggerating the performance value of the model [42]. Based on the SDMtools toolbox (http://www.sdmtoolbox.org/), we only keep one distributed data point in the range of 10 km × 10 km, reducing the occurrence position to a single point within the distance, and achieving the situation that the whole single grid meets the requirements.”

Comments 2: Meanwhile, based on the preliminary simulation results of MaxEnt model (Table 1), environmental factors with correlation ≥0.8 and low contribution rate were eliminated. I think the threshold is higher it should be less.

Response 2: Thank you very much for your opinions. It is very important for the selection of specific parameters in the modeling. However, in this study, 0.8 is selected as the threshold, mainly based on the specific prediction performance of the model and the research object. We explain as follows:

“When removing collinear environmental factors, it is necessary to consider the potential deviation or limitation of the data set to ensure the accuracy and reliability of the model. There are some ways to gain a deeper understanding of the potential deviations or limitations of data sets and determine which one to keep or delete. First, we keep environmental factors which are ecologically reasonable and related to the habitat needs of T. nanus. Second, we make sure that eliminating environmental factors will not reduce the prediction performance of the model. Third, based on target species, in the study of endangered plants is mostly set 0.8 as the benchmark [43, 46]. The strict conditions are necessary for their narrow distribution area and small population size”. (Line: 171-176)

Similar threshold chosen in referenced articles:

Ye, X. Z.; Zhang, M. Z.; Lai, W. F.; Yang, M. M.; Fan, H. H.; Zhang, G. F.; Chen, S. P.; Liu, B. Prediction of potential suitable distribution of Phoebe bournei based on MaxEnt optimization model. Acta Ecol. Sin. 2021, 41, 8135–8144. doi: https://doi.org/10.5846/stxb202007131822

Zhan, P.; Wang, F. Y.; Xia, P. G.; Zhao, G. H.; Wei, M. T.; Wei, F. G.; Han, R. L. Assessment of suitable cultivation region for Panax notoginseng under different climatic conditions using MaxEnt model and high-performance liquid chromatography in China. Ind. Crop. Prod. 2022, 176, 114416. doi: https://doi.org/10.1016/j.indcrop.2021.114416

Comments 3: Although MaxEnt is a well-known model for species distribution modeling, briefly discussing why it was chosen over other models, based on your specific research questions and data characteristics.

Response 3: Yes, when considering the choice of MaxEnt model, we need to consider the particularity of the research problem and the characteristics of the data. This part can be improved in Line 183-202 in manuscript-clean. The text is:

“The particularity of the research problem: (1) MaxEnt model has a good prediction ability in the study of endangered species [23, 43]. (2) MaxEnt model can be used to predict the current state of species distribution, future scenarios, niche expansion and other different forecasting purposes. This study not only predicts the current distribution, but also the past and future. Data characteristics: (1) in this study, the T. nanus has small population, and the data amount is not much. The MaxEnt model has strong processing ability for small sample datasets, so it is suitable for the case with small sample size [14]. (2) Environmental factors such as isothermality and land use type are discrete variables, while precipitation is continuous variables. MaxEnt model can integrate various types of environmental factor data, including continuous variables and classified variables, and can include the interaction of different types of variables [47]. (3) T. nanus is a rare and endangered species with narrow distribution. Reliable prediction could be produced by MaxEnt model for discrete species distribution data, so it is suitable for dealing with uneven distribution or locally rare species [23]. Comparing with other models, MaxEnt model has the best explanatory power and the most accurate prediction performance [18–19], a concise mathematical definition [14], capacity of avoiding sampling bias problem [51], and suitability for the conservation planning algorithm [42]. Therefore, we choose this model to predict the suitable area of endangered plant T. nanus.”

Comments 4: A Mentioning the spatial resolution is good, but discussing the implications of this choice on the accuracy and granularity of your predictions would be insightful. For instance, how does the 1km resolution affect the modeling of a species with small, scattered populations?

Response 4: Thank you, this suggestion has been followed as:

“We think that the Spatial resolution should be selected according to the ecological and biological characteristics of plant species. Variables should affect the distribution of species on relevant scales, which depends on the geographical scope and granularity of modeling tasks [47]. On one hand, if the spatial resolution of environmental factors is too coarse, some environmental factors may be integrated, which is unable to reflect the real influ-ences on T. nanus. On the other hand, constraining by suitable habitats and seed disper-sal, T. nanus has small and scattered populations in its geographical range [39]. Envi-ronmental variation in the geographical range could be well indicated by 1 km resolu-tion variables. And 1 km resolution bioclimatic data of WorldClim is accessible and is a rational choice. Therefore, we choose the spatial resolution of all data as 1 km.”

It can be found in Line 159-170 in manuscript-clean.

Comments 5: For readers less familiar with the specific models and statistical methods used, a brief explanation of terms like "regularized multipliers" and "feature types" in the context of MaxEnt modeling could improve the accessibility of your methodology.

Response 5: Thanks for your suggestion. In MaxEnt model, regularization multiplier (RM) is an adjustment parameter to balance the fitting degree and complexity of the model. The greater the regularization multiplier, the greater the penalty for the complexity of the model, so it is more inclined to choose a simple model; the smaller the regularization multiplier, the smaller the penalty for model complexity, and the more prone to over-fitting. Feature combination (FC) refers to combining different environmental factors to form new features, so as to increase the model’s ability to explain species distribution. Through feature combination, the interaction and nonlinear relationship between environmental factors can be better captured, thus improving the fitting ability and prediction performance of the model. In practical application, it is very important to choose the appropriate regularization multiplier and feature combination, which needs to be determined by cross-validation and other methods.

Additional explanation is input in the text, in Line 203-214 in manuscript-clean.

Results

Comments 1: While the results are robust, further clarifying how environmental factors were weighted or integrated into the model could provide deeper insights into their relative importance and interactions.

Response 1: Thank you very much for your suggestion. We agree and have revised it Line 272-294 in manuscript-clean. The specific revision is as follows:

“MaxEnt model weights or integrates environmental factors into the model mainly through the following steps [14]: (1) select of environmental factors that may affect species distribution based on ecological knowledge, field investigation or literature review. (2) Obtain GIS data of selected environmental factors, which are usually provided in grid or vector format and cover the spatial scope of the study area. (3) Weight environmental factors based on the correlation between environmental factors and known species distribution. MaxEnt model can track the environmental variables with high contribution rate, and gradually modify a single factor to increase the gain value, and assign the gain value to the environmental variables on which the factor depends, which is expressed as a percentage [68]. The importance of permutation is determined by the random substitution value of environmental variables at training points and the decrease of AUC value. The decrease of AUC value indicates that the variable depends on the model [69]. Jackknife test uses one variable in turn or excludes one variable in turn to build models, and compares the differences of training gain, test gain and AUC value between model to evaluate the importance of environmental variables [70]. (4) Model training: the MaxEnt model uses the known species distribution points and the corresponding environmental factor data for training. In the training process, the model will learn the relationship between environmental factors and species distribution, and optimize the model parameters to fit the known data to the greatest extent. (5) Predicting species distribution: The trained MaxEnt model can be used to predict species distribution in unknown places. In the process of prediction, the model will calculate the suitability of each site for species survival according to the data of environmental factors and the learned weights”.

Comments 2: Expanding on the ecological implications of the dominant environmental factors and habitat shifts could enrich the discussion. For instance, how do these shifts correlate with broader ecological trends in Yunnan or similar regions?

Response 2: We agree on your concerns. The dominant environmental factors and habitat shifts are expanded and enriched in discussion part in Line 464-478 in manuscript-clean. The specific contents are as follows:

“The change of environmental factors will not only have a great impact on an endangered species, but also enlarge the loss of biodiversity. Because the survival of plants is not isolated, but in a certain interdependent relationship with other species [93]. The disappearance of a species often leads to the coexistence crisis of 10–30 species [94]. Therefore, the changes in temperature and precipitation caused by climate change have far greater impact on an endangered species itself, and may change the structure and function of the ecosystem [95]. Global warming usually leads to the advance of plant phenology [96]. However, the same period of precipitation and heat in the distribution area of T. nanus is an important factor for its growth [33], and the increase of temperature will increase the occurrence of drought [97], which will most likely cause the growth of T. nanus to be limited by rainfall. The growth of T. nanus is slow and its regeneration ability is relative weak [33]. When its own habitat is sup-pressed, its ability to expand and compete for niche is weaken. Meanwhile, climate changes may lead to ecosystems degradation, such as flash floods, droughts and forest fires, which further aggravate the extinction possibility of endangered plants”.

Comments 3: While the paper mentions different climate scenarios, a more detailed discussion on how each scenario could specifically impact conservation planning and management strategies for T. nanus would be beneficial.

Response 3: Thanks for your suggestion. Effective measures to ensure the long-term survival and health of T. nanus is important in future climate change and should be specific. Conservation planning and management strategies impacted by each climate scenario are improved in the discussion. This part can be found in Line 527-547 in manuscript-clean. Our improvements are as follows:

“Therefore, according to the characteristics of different climate scenarios (SSP1-2.6, SSP2-4.5, SSP5-8.5) [105], we put forward different management strategies for the protection planning of T. nanus. SSP1-2.6 scenario is low-carbon emission and highly sustainable development. Under this scenario, climate change is effectively controlled and there may be a moderate trend of warmth and humidity. It has a relatively positive im-pact on the living environment of T. nanus. Conservation planning and management strategies should focus on maintaining the suitable area of T. nanus. SSP2-4.5 scenario is a moderate carbon emission and a moderate development path. Under this scenario, temperature will rise and precipitation will change, resulting in a certain impact on T. nanus. Protection strategies need to actively respond to the climate change impact, such as strengthening the continuous monitoring of T. nanus growth and the management of protected areas. Scenario SSP5-8.5 is a development path with high carbon emissions and high inequality. Under this scenario, greenhouse gas emissions are high and there are many extreme events, which may lead to serious climate change and environmental damage. It may pose a serious threat on T. nanus. Protection enforcement and implementing emergency ecological restoration are necessary for this situation. Multiple measurements should be appliable, such as in-situ and ex-situ conservation, genetic resources recovery, intense monitoring, etc. T. nanus’ protection planning and management strategies need to be adjusted and customized according to climate scenarios. It is important to take effective measures in time to ensure the long-term survival and health of T. nanus populations and its habitat”.

Discussion

Comments 1: While the discussion on the main environmental factors and model predictions is robust, expanding on the broader ecological implications of these changes could provide a more comprehensive view.

Response 1: Thanks for your suggestion. Broader ecological implications of climate changes on endangered plants are input in the discussion. The details can be found in Line 518-526 in manuscript-clean. The specific contents in the revised manuscript are as follows:

“The change of the suitable area of T. nanus may also have a certain impact on the ecosystem. First of all, although this species exists as a companion species in the com-munity, other plant and animal species must have certain dependence on its existence [93]. For example, T. nanus provide food for some birds and ferrets in local ecosystem (field observation), and the change of its suitable area may lead to the destruction of food chain or ecological network. Secondly, T. nanus extinction caused by habitat shifts will reduce the species richness of ecosystems. Finally, the change of the suitable area of endangered species may affect local community composition and unique ecosystem function”.

Comments 2: The call for more protected areas and corridors is well-justified, but providing more specific recommendations or potential models for these conservation strategies could enhance the section. This might include examples of successful conservation efforts for similar species or habitat types, which could serve as a blueprint for T. nanus.

Response 2: Thanks for your suggestion. Relative part has been improved. It is in Line 609-614 in manuscript-clean. The revised contents are as follows:

“For example, in-situ and ex-situ conservation of the extremely small population species Salvia daiguii saved the populations [125]. The endangered species Saraca asoca has greatly improved its growth under the nature reserve [126]. Yellowstone-Yukon Ecological Corridor promotes the migration of various animals and plants and protects the habitats of many endangered species, including gray wolves and bears [127]. Nature reserves and ecological corridors provide good habitats and interconnected channels for organisms in different regions, which enhance biodiversity and ecosystem health. At the same time, they will also promote the animal dispersal of T. nanus’ seeds to enhance gene flow and population regeneration”.

Comments 3: Discussing potential adaptation and mitigation strategies for the impacts of climate change on T. nanus would be a valuable addition.

Response 3: Thanks for your suggestion. Relative part has been improved. The potential adaptation and mitigation strategy of climate change on the impact of T. nanus will be a valuable supplement to this study. You can see it in Line 602-608 and 619-634 in manuscript-clean. The specific supplementary contents are as follows:

“Plants adapts to climate change based on its biological properties [123]. T. nanus has a long life span and slow regeneration ability, it takes about 10 years that from seed germination to seedling begin flowering and fruiting [33]. Therefore, it is difficult for T. nanus to expand its actual niche under rapid climate change. The growth of T. nanus seedlings needs shadow habitat. Commonly, it is a companion species in pinewood or ever-green board-leaved forest [32], with the reduction in forest area [124], the regeneration of T. nanus seedlings would be influenced”.

Comments 4: While the discussion acknowledges human impacts on T. nanus, elaborating on strategies for engaging local communities in conservation efforts could add depth. Community-based conservation strategies could be particularly relevant for species affected by habitat fragmentation and land-use changes.

Response 4: Thank you, this suggestion has been followed in Line 623-634 in manuscript-clean. The contents in the revised manuscript are as follows:

“In addition, local people may have rich knowledge in the distribution and growth habits of this species, and the forestry department (in charge endangered plants protection) needs to cooperate with the local communities to jointly conduct the protection plan. Strengthening local people’s awareness on T. nanus’ protection will reduce anthropogenic threats, such as picking flowers and fruits, stock grazing. The implement of T. nanus’ protection will facilitate afforestation and improve habitat quality which benefit local communities. Sustainable T. nanus protection should be enhanced by reasonable resource exploration, and more researches are needed. T. nanus has a potential value in gardening, appropriate artificial plantation and breeding will promote wild population protection. Researchers should carry out research on germplasm resources and promote genetic improvement. The state and local governments and local stakeholders should work together to facilitate the protection in a sustainable way”.

Conclusion

Comments 1: While you suggest reducing human interference and protecting indigenous plant communities, elaborating on specific measures (e.g., sustainable land-use practices, and community engagement in conservation) could provide a more actionable framework for stakeholders.

Response 1: We sincerely appreciate your valuable comments. As you suggested, we think it is necessary to provide specific and operable protection measures. We have improved in Line 618-634 in manuscript-clean. The details are as follows:

“We strongly suggest that more small protected sites should be established, as well as corridors for population connectivity to ensure the proliferation and recovery of T. nanus under climate change. In addition, the local community engagement in conservation should be encouraged, such as joint-management, mass education, sustainable resource exploration, etc”.

Comments 2: Addressing the uncertainties in climate change projections and their implications for T. nanus could lead to discussions on adaptive management strategies.

Response 2: Thanks for your suggestion. Adaptive management strategies for the uncertainty in climate change projections and their implications for T. nanus are improved in the discussion. This can be found in Line 604-608 in manuscript-clean.

Comments 3: Expanding on how global carbon neutrality efforts could mitigate climate change impacts on T. nanus would provide a hopeful perspective on the potential for policy actions to influence conservation outcomes positively.

Response 3: Yes. Positive conservation of T. nanus relied on global carbon neutrality efforts. At present, on the basis of striving for the goal of “double carbon”, global carbon neutrality needs the joint efforts and policy support of the international community to reduce greenhouse gas emissions. Under this guarantee, less climate change will reduce the habitat shifts of T. nanus and direct influences on its growth. This part can be found in Line 644-652 in manuscript-clean.

Reviewer 3 Report

Comments and Suggestions for Authors

Here are some suggested revisions for your manuscript: 

Introduction: Emphasize the significant uses of T. annus, such as its role in medicine, and argue for the importance of protecting this species due to its unique properties and potential benefits to society. Highlight the urgency of saving this plant species from extinction. 

Abstract: Briefly mention the significant uses of T. annus and the importance of protecting it to maintain biodiversity. Summarize the urgency of conservation efforts. 

References: Ensure that references are cited correctly and clearly. Avoid using numbers that could be confused with other elements in the text. Consider using a consistent citation format throughout the manuscript. 

Human Activities: Expand on human activities impacting T. annus, such as wind power development zones, and include other relevant activities that contribute to the species' decline. 

Discussion/Conclusion: Address management strategies for conserving T. annus and maintaining biodiversity. Discuss how conservation efforts can be implemented effectively. 

Additional Factors: Besides climate change and human activities, discuss other potential reasons for the rarity and endangerment of T. annus, such as habitat loss, invasive species, and pollution.

Overall, the manuscript provides valuable insights into the conservation of T. annus. However, improvements in emphasizing its significant uses, addressing human activities, and discussing additional factors contributing to its rarity and endangerment would enhance the manuscript's impact and clarity.

Author Response

Response to Reviewer 3’s Comments

Comments 1: Introduction: Emphasize the significant uses of T. annus, such as its role in medicine, and argue for the importance of protecting this species due to its unique properties and potential benefits to society. Highlight the urgency of saving this plant species from extinction.

Response 1: Thanks for your suggestion. T. nanus is a national second-class rare and endangered plant in China. The flowers and fruits of this species are edible or medicinal. Its fruits are food for birds and small vertebrates. It is only shrub species of Trachycarpus plants in China, and has a close phylogenetic relationship to Chamaerops humilis which occurs on Mediterranean shore. This might infer the ancient geological history of Tethys Sea. So, it has high ecological, scientific and medicinal values. At present, this species is on the verge of extinction for its small population, and threated by human disturbance and alien invasive plants, as well as climate change. Related contents have been improved in Line 81-87 in manuscript-clean.

Comments 2: Abstract: Briefly mention the significant uses of T. annus and the importance of protecting it to maintain biodiversity. Summarize the urgency of conservation efforts.

Response 2: Thank you for your suggestions. The revised details are as follows:

T. nanus is a national second-class rare and endangered plant in China, and endemic to Yunnan. Its flower is medically used for curing nephropathy, the seed for dizziness and headache, and the leaf sheath can be used as an astringent hemostatic. With narrow distribution range and extremely small population, it is facing much more risks under global climate change.” Relative text is in Line 20-22 in manuscript-clean.

Comments 3: References: Ensure that references are cited correctly and clearly. Avoid using numbers that could be confused with other elements in the text. Consider using a consistent citation format throughout the manuscript.

Response 3: Thank you sincerely for pointing out our previous mistake, and it has been corrected in revised manuscript.

Comments 4: Human Activities: Expand on human activities impacting T. annus, such as wind power development zones, and include other relevant activities that contribute to the species' decline.

Response 4: Thanks for your suggestion. Based on former researches and field observation, human activities such as deforestation for agricultural practice, leaf-cutting for brooms, goat grazing, road and wind power farm construction. Brief discussion is improved in relative context, Lines 81-93 in manuscript-clean. The specific contents are as follows:

“The development zones with good wind energy resources in Yunnan Province cover most areas where the T. nanus is distributed [34]. Wind power farm construction put threats on its habitats [35–36]. In addition, livestock overgrazing (Figure S1–B, C) [29] and road construction [28] are also human activities that threaten T. nanus. According to the field investigation and literature, the invasive species, Ageratina adenophora has occupied a large number of T. nanus’ habitats (Figure S1–D), seriously affecting the survival and regeneration of its seedlings [37]”.

Comments 5: Discussion/Conclusion: Address management strategies for conserving T. annus and maintaining biodiversity. Discuss how conservation efforts can be implemented effectively.

Response 5: Thanks for your suggestion. Effective measures to ensure the long-term survival and health of T. nanus is important in future climate change and should be specific. Conservation planning and management strategies impacted by each climate scenario are improved in the discussion. This part is in Line 618-634 in manuscript-clean.

Comments 6: Additional Factors: Besides climate change and human activities, discuss other potential reasons for the rarity and endangerment of T. annus, such as habitat loss, invasive species, and pollution.

Response 6: Thank you. We focus on climate change impact on T. annus distribution and habitat loss, our present work is unable support further analysis for these factors. In addition, we improved related text in Line 79-87 in manuscript-clean.

Round 2

Reviewer 2 Report

Comments and Suggestions for Authors

I am pleased to say that I accept your paper. Congratulations on reaching this milestone.